# Threats and Vulnerabilities for the Globally Distributed Olive Ridley (*Lepidochelys olivacea*) Sea Turtle: A Historical and Current Status Evaluation

**DOI:** 10.3390/ani12141837

**Published:** 2022-07-19

**Authors:** Lenin Cáceres-Farias, Eduardo Reséndiz, Joelly Espinoza, Helena Fernández-Sanz, Alonzo Alfaro-Núñez

**Affiliations:** 1AquaCEAL Corporation, Urb. Las Palmeras, Ave. Capitán Byron Palacios & General Quisquis, Santo Domingo de los Colorados 230101, Ecuador; caceres_lenin@hotmail.com; 2Grupo de Investigación en Biología y Cultivo de Moluscos, Departamento de Acuicultura, Pesca y Recursos Naturales Renovables, Facultad de Ciencias Veterinarias, Universidad Técnica de Manabí, Bahía de Caráquez, Manabí 131101, Ecuador; 3Departamento Académico de Ciencias Marinas y Costeras, Universidad Autónoma de Baja California Sur (UABCS), La Paz 23080, Mexico; jresendiz@uabcs.mx (E.R.); joelly.espinoza@uabc.edu.mx (J.E.); helena.fdezsanz@gmail.com (H.F.-S.); 4Health Assessments in Sea Turtles from BCS, La Paz 23085, Mexico; 5Section for Evolutionary Genomics, GLOBE Institute, University of Copenhagen, Øster Farimagsgade 5, 1353 Copenhagen K, Denmark; 6Department of Clinical Biochemistry, Naestved Hospital, Ringstedgade 57a, 4700 Naestved, Denmark

**Keywords:** endangered species, marine turtles, population decline, conservations strategies, climate change

## Abstract

**Simple Summary:**

Showing one of the most extraordinary phenomena in nature, with thousands or hundreds of thousands of females coming out of the water to nest in a highly synchronized event, specific to very unique and yet birth-determined beach remote locations, the olive ridley is the second smallest and the most cosmopolitan of the seven sea turtles species of the world. Nevertheless, olive ridley populations have dramatically decreased during the last 50 years, nearly entirely caused by anthropogenic but also environmental and climatic pressures. In the following pages, we attempted to provide a very broad description of the main biological and ecological features of this species from a historical perspective to their current status. We identified and listed the main threats to which olive ridleys are currently exposed. By the end of the document, we listed a number of general recommendations to help the protection and conservation of olive ridleys.

**Abstract:**

The olive ridley (*Lepidochelys olivacea*) is the most abundant of all seven sea turtles, found across the tropical regions of the Atlantic, Pacific, and Indian Oceans in over 80 different countries all around the globe. Despite being the most common and widely distributed sea turtle, olive ridley populations have been declining substantially for decades. Worldwide, olive ridleys have experienced a 30–50% decline, putting their populations at risk and being considered an Endangered Species by the IUCN. Natural habitat degradation, pollution, bycatch, climate change, predation by humans and animals, infectious diseases and illegal trade are the most notorious threats to explain olive ridley populations rapidly decline. The present review assesses the numerous dangers that the olive ridley turtle has historically faced and currently faces. To preserve olive ridleys, stronger conservation initiatives and strategies must continue to be undertaken. Policies and law enforcement for the protection of natural environments and reduction in the effects of climate change should be implemented worldwide to protect this turtle species.

## 1. Introduction

Sea turtles are considered to be necessary for a healthy ocean, as they have a direct impact on other species. They contribute to the primary interactions in the evolution, structure and dynamics of multiple marine ecosystems. As such, sea turtles play multiple roles as prey, consumers, hunters, competitors, as well as carriers or hosts for other organisms. Therefore, turtles are widely considered an important part of interspecific interactions in marine ecosystems [1], and they are considered to be a potential ecosystem health bioindicator [2]. Nevertheless, sea turtle populations have been declining for decades all over the world. This decline has been so serious that all turtle species have been classified as vulnerable, endangered, critically endangered, near threatened, or data deficient [3].

The olive ridley (*Lepidochelys olivacea*) sea turtle is considered to be one of the smallest and most abundant of all seven turtle species of the world. Yet, it is threatened and endangered in many areas, with smaller populations in the western Pacific displaying some of the most dramatic declines [4,5]. Olive ridleys were formerly thought to be resilient to overexploitation because of their abundance. This was mistakenly far from the reality. Indeed, in the period from the 1960s through the 1980s, large-scale industrial extraction caused such frightening population crashes in several rookeries, particularly in Mexico [6], that the species was classified as endangered on the International Union for the Conservation of Nature (IUCN) Red List of Threatened Species [5]. 

The olive ridley gets its name from the olive green colour of its heart-shaped shell, morphological features shared only with the smallest Kemp’s ridley (*Lepidochelys kempii*) turtle; however, these features clearly differentiate them from the other five marine species. Olive ridleys have a circumtropical worldwide distribution and are widely known for their “arribadas” (Spanish for “arrivals”), or mass nesting events, which only happen in a few unique locations in Costa Rica, India, and Mexico [7,8,9,10]. There is, however, less available information about the populations in the many other nesting sites across the world where this turtle nests in low density, often on secluded beaches [7,11,12]. This is especially true in the western Pacific and Southeast Asia, where there is little knowledge on their distribution and abundance [9,11,13]. Olive ridleys used to be common in Myanmar, Thailand, and Peninsular Malaysia, but decades of intensive egg harvesting have reduced and, in some cases, entirely depleted populations in most of these countries [14]. 

Olive ridleys show a great behavioural plasticity across the world. They can migrate through oceanic waters and feed on surface fauna, or remain in shallow coastal waters where they feed on invertebrates [15,16,17]. The nomadic behaviour of many olive ridley turtle populations represent a successful biological strategy for the opportunistic search for prey, which is patchily distributed. This flexibility might allow them to cope with unpredictable changes in dynamic ecosystems, and could help explain why olive ridleys are the most abundant of all sea turtles [5,18]. However, because of their slow intrinsic growth rate and anthropogenic pressures, olive ridley turtles are vulnerable, and many populations have decreased Coastal development and commercial fishing activities continue to degrade, change, and destroy natural conditions at nesting and feeding grounds, endangering the long-term existence of many olive ridley rookeries worldwide [4,19]. Along the seaside, olive ridleys face a number of dangers. As such, *Casuarina* plantations, beach erosion, construction of touristic complexes, artificial illumination, and predation of eggs and hatchlings are factors largely damaging olive ridleys ecosystems through the loss of their natural nesting habitats [20]. 

In oceanic waters, olive ridleys are exposed to a variety of dangers such as harmful algal blooms, climate change, bycatch, directed fishing nets capture, marine litter, and contamination, which can be fatal and a potential cause of death [21,22,23,24]. Assessing a species’ risk of extinction is difficult; it requires a thorough understanding of global trends spanning generations [25]. The loss changes in population size of olive ridleys drive their worldwide status. Despite the fact that monitoring arribada sizes has proven to be difficult, local initiatives have provided solid data for status assessments. Decades of conservation efforts, including nesting beach protection and rules prohibiting sea turtle commerce and direct capture on land and at sea, in particularly across the Americas (e.g., Costa Rica and Mexico), have resulted in highly promising current trend statistics [25]. Over the last two decades, most arribada rookeries have had positive or stable trends in the Americas; nonetheless, populations across Asia are still in rapidly decline [5]. However, in comparison to the past, the population is fast dwindling [26]. Some efforts have been made worldwide to safeguard the olive ridley turtle. In this document, we attempted to provide the past and present status of the olive ridley turtle and the risks it currently faces. We also look at the existing efforts to further protect this turtle species. Finally, we list some general perspectives for the conservation of the olive ridley turtle. 

## 2. General Characteristics

The olive ridley turtle (*Lepidochelys olivacea*), often known as the Pacific ridley, belongs to the Cheloniidae family of turtles. The olive ridley takes its name from the colour of its heart-shaped shell, skin, or carapace, which resembles an olive green. The olive ridley is the world’s second-smallest species [27] of sea turtle. The olive ridley is closely related to its smaller sister the Kemp’s ridley, with the main behavioural difference being that they exclusively reside in warmer waters, endemic to and uniquely found in the Gulf of Mexico. This turtle is well known for its arribadas, or mass nesting sites, when thousands of female turtles congregate on the same beach to lay eggs [4]. Worldwide, with a yearly estimated population of at least 800,000 nesting females, the olive ridley turtle is considered the most prolific sea turtle on the planet [28]. 

### 2.1. Global Distribution

The olive ridley turtle is widely distributed throughout the tropical and subtropical waters of the Pacific, Atlantic, and Indian Oceans (Figure 1), predominantly in the Indo-Pacific region [29,30]. In the Pacific, this species has been reported from northern Mexico to Chile, including central Pacific islands [18,31,32,33], while in the Atlantic, there are reports of olive ridleys from the Azores Islands, in the North Atlantic, to Uruguay [34,35].

Olive ridley turtles have a highly flexible migratory lifestyle and use a wide variety of habitats [18]. In the Pacific region, they spend most of their life in the oceanic zone, whereas Atlantic turtles are commonly found in neritic areas [15,16,17]. Between pelagic feeding and coastal breeding areas, olive ridleys traverse from a few to thousands of kilometres, depending on the ecological conditions of these sites [18,37,38,39]. Adult olive ridleys have been observed nearly 4000 km from land in the Pacific, according to fishermen observations [40]. 

The synchronized event of massive numbers of nesting females (above 1000 females) on a single location over a short period of time, or arribadas, is unique to olive ridley sea turtles [41]. However, similar pattern behaviour has also been observed during the large green turtle (*Chelonia mydas*) nesting seasons at Tortuguero, Costa Rica [42]. There are currently at least five main sites producing more than 100,000 nests per year, and 8–15 minor sites producing 10,000–100,000 nests per year [5] around the world (see Figure 1). The Pacific coasts of Mexico and Central America, the Atlantic coasts of South America, the west coast of Africa, and the South and Southeast coast of Asia all have beaches that house hundreds to thousands of nests per year [43,44]. However, the main arribada nesting areas are located on the Pacific coast of Mexico, in Costa Rica, and in northeast India [45,46,47]. There are also several beaches reported as olive ridley solitary nesting spots in these regions [25,48], as well as in the Central African Atlantic Coast [26,49], Australia [50], French Guiana and Brazil in the western Atlantic [17,39], South American Pacific [51], South and Southeast of Asia [52], and on most island groupings throughout the tropics [5,53].

Historically, Mexico has had the largest arribada sites on the planet, with the greatest occurring on Playa La Escobilla [5]. However, between the 1960s and 1990s, the sea turtle fishery in conjunction with the shrimp fishery caused the overexploitation of the olive ridley turtles [6]. During this period, tens of thousands of olive ridleys were slaughtered annually in Mexico to supply hides as a substitute for rare crocodile leather to a developing international commerce [54]. The famed turtle butcher in San Agustinillo, Oaxaca, was shut down in the 1980s after a global uproar about the declining abundance of turtles and the collapse of arribadas at Mismaloya, Tlacoyunque, and Chacahua. In 1990, Mexico enacted a permanent ban on the exploitation of sea turtles [55]. After that, nesting at Playa Escobilla increased fivefold, from around 200,000 nests per year in the 1990s to over 1 million by 2000; the number is now stable, with about nine arribada events each year. Every year, almost 1 million nests are found on the adjacent beach of Morro Ayuta [56]. 

Arribadas are known to exist in three countries in Central America: Panama, Nicaragua and Costa Rica (Figure 1) with the largest arribada events in the region. Panama has the smallest population of the three, but Nicaragua has considerable aggregations near La Flor and Chacocente [57,58]. Costa Rica has regular arribada rookeries in Ostional (the largest arribada in the region and third largest in the world) and Nancite, and two new arribada rookeries are forming at Corozalito and Camaronal [59]. Nancite’s arribada is a peculiar case. The arribada is located on a small beach within Santa Rosa National Park, so it is largely free of the manmade hazards that generally harm turtles, yet there has been a 90% decline in nesting abundance there since the early 1970s [60]. The large numbers of turtles nesting on top of each other on this tiny beach (less than 1 km long) are likely to have resulted in a high number of broken eggs and a high microbial load across the entire beach [61]. Because of microbial activity, a lack of oxygen might suffocate developing embryos, resulting in low hatching success [62]. The ensuing poor population recruitment over a long period of time may have caused the collapse at Nancite. On the other hand, the Ostional nesting site also represents a unique set of very different conditions from the ones describe above at Nancite. Since 1987, a unique and controversial community conservation program has been operating in Ostional, which aimed to generate income exclusively for the community by promoting the protection of olive ridleys through the legal harvest and trade of eggs [63]. Despite the importance of this nesting site, only a few studies have evaluated the population trend in the area and its relationship with the program’s activities. Long-term studies suggest that from the 1980s to the early 2000s, the population remained stable or increasing despite egg harvest [64,65], while short-term studies have proposed a possible decrease in the population trend [66]. Nevertheless, the causes of these oscillations of population size still remain unknown.

The major mass nesting sites in India are located in Odisha, on the east coast. In this region, usually during the dry season, one or two significant arribadas occur there every year [67]. The beaches along this region are made up of sand bars that erode and accumulate over time. However, that have changed dramatically in recent decades, and it could potentially be affecting these mass-nesting events. During the 1970s, tens of thousands of olive ridleys were captured in Odisha and shipped to Kolkala (formerly Calcutta), the capital of India’s West Bengal state, where the meat was widely consumed. Concerns raised by international and local conservationists led to the passage of wildlife regulations putting an end to this practice [5]. In the 1980s, mass nesting was recorded near the mouth of the Devi River, but no arribadas have been seen there since 1997. Over the last 20 years, the topography of Rushikulya, the southernmost of Odisha’s mass nesting locations, has remained largely unchanged. As a result, in the 2010s, arribada nesting increased to more than 200,000 nests in a single event [5]. 

The locations and sizes of arribada rookeries are changing rapidly [45]. For instance, nesting began in the late 1990s in Ixtapilla, Mexico, and by 2009, this arribada had grown to over 200,000 nests every year. Nesting has expanded from around 1000 nests per year in 2008 to over 47,000 nests in 2019 in Costa Rica’s two new arribada beaches, Corozalito and Camaronal [5,68]. In the early 2010s, a new rookery formed in India’s Andaman archipelago was reported that now supports 5000–10,000 nests per year. On rare occasions, several beaches in those regions, as well as in Suriname and French Guiana, have small arribadas with a few hundred nests [5].

Although sea turtle scientists previously assumed that the extinction of arribadas was solely due to human activity, there could be other factors at work [69]. The changing nature of the beaches, together with the large swings in nest numbers over short times, and the detrimental influence on hatching rates caused by the build-up of organic debris from broken eggs, may create oscillations in the presence and size of arribadas, which rely on environmental conditions, as Nancite rookeries have historically proven [70]. It has been observed that solitary turtles and most arribada nesters appear to prefer beaches near river mouths. Seasonal flooding removes organic accumulation cleaning the beaches, which may be the best nesting places, allowing turtle populations to thrive in the long run. Nevertheless, it is still a mystery how and why arribadas are born, expand, contract and collapse [68]. 

### 2.2. Habitat, Food and Feeding Behaviour

Olive ridleys are generally found in coastal bays and estuaries, but certain portions of its range might be highly oceanic. The majority of observations occur within 15 km of mainland coasts in protected, somewhat shallow marine waters of approximately 22 to 55 m deep [71]. Every year, olive ridley travel hundreds or even thousands of kilometres to join the arribada, when vast groups of females return to the beaches where they were previously believed to be born to lay their eggs [20]. 

During the non-reproductive period, this species uses a variety of habitats for feeding, showing a preference for sites with high levels of biological productivity on the coast, continental shelf, and continental slope [72]. They can use diverse feeding strategies, either foraging in open waters off the coast, descending to depths of 150 m to feed on bottom-dwelling crustaceans, meandering over neritic areas using various feeding grounds, or settling directly at specific feeding sites at river mouths [15,72]. 

Olive ridleys present a generalist feeding behaviour [40]. They have strong jaws that enable them to crush their food. The olive ridley is primarily a carnivore, especially in its early phases of development, and can feed on pelagic or benthic prey [73]. They are also omnivorous, which means they eat both plants and animals. Marine invertebrates found in shallow seawaters or estuary settings represent the most regular type of diet for olive ridleys [74]. Thus, jellyfish, tunicates, sea urchins, bryozoans, bivalves, snails, shrimp, crabs, rock lobsters, echinoderms and sipunculid worms are all common prey [75,76]. Consumption of jellyfish, as well as adult and juvenile fish (e.g., Sphoeroides) and fish eggs, may indicate a pelagic feeding behaviour also observed in this species [71,73]. In regions where there are no other food sources available, the olive ridley is reported to eat filamentous algae (seaweed). Studies in captivity have revealed that this species is capable of cannibalism, in particular of small hatchlings [77]. Because the olive ridley turtle eats a wide variety of foods, marine litter such as plastic bags and styrofoam poses a significant danger to these animals.

### 2.3. Ecological Rolls 

Sea turtles are an integral element of the planet’s food web chain, and they play a key role in keeping the world’s oceans healthy. They have the ability to manage a wide range of other creatures just by ingesting them [78,79]. Olive ridley turtles eat invertebrates and are thought to play a vital role in both open coastal and marine ecosystems [30]. Additionally, as consumers, olive ridleys maintain the balance of the food web and favour the control of many populations [1]. Olive ridleys play also an essential role in nutrient transport, disseminating large amounts of nutrients from feeding areas to nutrient-depleted coastal ecosystems near nesting beaches [79,80]. 

Unhatched eggs and empty eggshells left at the sand in nests on beaches act as a fertilizer for coastal vegetation, providing nourishment for plant growth and helping to stabilize the coastline while also providing food for a range of plant-eating animals [79,81], as well as invertebrates and microorganism living in the sand. Sea turtles play a vital role in producing and sustaining diversity in the world’s waters by transferring creatures that reside on reefs, seagrass meadows, and the open ocean [82]. In order to ensure healthy marine ecosystems, and given the range of functions sea turtles have, we must sustain, protect and rebuild their populations.

### 2.4. Source of Food for Humans

There are centuries of records reporting the use and consumption of several species of sea turtles, olive ridleys within them. Sea turtle eggs are regarded as a delicacy all over the world. Because egg harvesting has the potential to boost local economies, numerous communities have experimented with the novel practice of allowing a sustainable (legal) egg harvest [9]. As mentioned above, it has been documented that the legal egg harvest in Ostional, Costa Rica, is both biologically and economically viable, despite the unpopularity of such practice by conservationists. Local people have been able to harvest and sell roughly three million eggs each year since egg harvesting became legal in 1987. They are allowed to collect eggs for the first 36 h of the nesting cycle since later nesting females would destroy the big majority of these eggs. Over 27 million eggs are left unharvested, and villagers have played a significant role in preserving these nests from predators (such as snakes and birds), resulting in higher hatching success [66,77]. In other areas where coastal communities and sea turtles coexist, residents also find benefit from working in community-based conservation programs, where they carry out activities such as nesting beach patrolling, nest collection, and hatchling release. In fact, this strategy has been essential for the recovery of olive ridley populations in many nesting beaches around the world [83,84,85]. 

In addition to its ecological and livelihood importance, and the subsistence benefits associated with olive ridley turtles, historically, this species represents an important figure in a variety of traditional cultures as they have played a prominent role in medicine, religious beliefs, and spiritual values. Nevertheless, the olive ridley, like any other sea turtle, may present a nuisance to commercial fishermen, who frequently discover these turtles entangled in their nets [86]. 

### 2.5. Habitat and Food Source for Other Marine Organisms

Many marine creatures benefit from the presence of sea turtles. Barnacles, algae, and epibionts in general attach themselves to the turtle shell, which the turtle then carries around as a source of food for fish and shrimp. In fact, olive ridley turtles host a greater diversity of epibionts compared to other sea turtle species, and several fish species rely solely on epibionts found in this turtle to survive [79,87,88]. 

Sea turtles are a vital food source for other organisms, particularly in their early phases of development. Unhatched turtle eggs are reported to be dug up by ants, crabs, rats, raccoons, foxes, coyotes, feral cats, dogs, mongoose, and vultures; the egg yolks are a nutrient-rich source of food. A variety of sea birds, fish, and invertebrates feed on hatchlings, whereas several species of sharks and killer whales feed on juvenile and adult sea turtles [79,88]. 

### 2.6. Past and Current Status of Olive Ridley Population Decline

It has been estimated that only 1 to 8% of eggs laid during the arribadas hatch, as a direct result of illegal egg poaching, turtle hunting, and nest destruction by humans [89]. Between 1988 and 2008, the population of olive ridley turtles was estimated to have a worldwide decrease by 28 to 32%. The last estimate for this species made in 2013 indicates that although olive ridley turtles are the most common, their numbers have decreased by more than 30% globally [90], following the same trend from the previous decade. Because of the relatively few surviving breeding locations in the globe, these turtles are considered endangered. 

Prior to the age of mass exploitation, the population of Pacific Mexico was estimated to be at least 10 million. The IUCN Red List classified olive ridleys as vulnerable, and the Wildlife Protection Act of 1972 was further established to protect them. On 28 July 1978, the breeding colony in Mexico was designated as endangered in the United States (U.S.) [74]. Later, by 2004, the global population of annual nesting females had been lowered to around two million [91], and by 2008 it had been further reduced to approximately 852,550 individuals [4,9]. This meant that the global population dropped alarmingly in just one generation (i.e., 20 years) [5,90]. 

## 3. Vulnerabilities and Threats for the Olive Ridley Turtle 

### 3.1. Illegal trade and Consumption of Turtles and Eggs 

Unfortunately, sea turtle eggs are still considered an exotic dish or believed to be an aphrodisiac in many cultures and locations around the world (despite the fact that this is not supported by science). Additionally, turtle flesh (mainly from green turtles) is still consumed, despite the fact that both are outlawed in most nations [88]. As such, a combination of a long-term egg collecting and mass slaughter of adult females on nesting beaches are the primary causes of the olive ridley sea turtle’s worldwide decline. The arribada nesting behaviour brings females and nests together at the same time and in the same location, allowing for the collection of a large number of eggs for human consumption, mainly if nesting occurs on beaches close to where the human population density is high [46]. Egg harvesting for human food was a major issue, but this hazard has been reduced in several nations owing to laws prohibiting turtle killing and egg gathering [86]. 

### 3.2. Bycatch in Fishing Gear and Entanglement in Abandoned Fishing Gear 

Sea turtles bycatch (especially in shrimp trawl nets, longlines, and gill nets) is an issue of global concern [92]. Bycatch is defined as the unintentional capture of non-target species such as sea turtles, sharks, sea birds, whales, etc., in fishing gear, which can result in drowning or injuries that lead to death or debilitation (for example, eating hooks or flipper entrapment), this being a major hazard (Figure 2). Trawls, longlines, gillnet, purse seines, and pot fishing are the most common types of gear that result in olive ridley bycatch [4,77]. More than 100,000 olive ridley turtles were estimated to have died in Odisha, India, between 1993 and 2003 as a result of fishery-related practices [93]. However, in Costa Rica, an approximate estimate of nearly 700,000 olive ridley turtles were trapped by longline fisheries between 1999 and 2010, meaning that this activity may have been partly responsible for the decrease in the nesting population on arribada beaches during that period [94].

Entanglement in abandoned fishing nets or fishing gear debris can quickly kill sea turtles by drowning them or preventing them from fleeing predators or hunting [95]. Moreover, sea turtles have widely been reported to ingest plastic debris from fishing gear such as nets. They are also prone to getting tangled up in ghost gear (fishing nets discarded in the ocean). Foraging area locations can overlap with trawl fishery areas, increasing the possibility of turtles becoming entangled in longlines, shrimp nets, and bomb fishing, among other things [50,88]. The Maldives islands have reported 131 entangled sea turtles during a period of 5 years in trawl or gill nets, and olive ridley turtles appeared to be the most vulnerable accounting for 97% of the incidents [19]. Unfortunately, the accumulation of plastics debris from fishing nets at crucial nesting sites puts both adult females and hatchlings turtles at risk of becoming entangled in it, preventing them from reaching the sea [95]. 

### 3.3. Attack by Predators and Predation of Eggs and Hatchlings 

Wild (e.g., racoons, coyotes and jaguars) and semi-domesticated (e.g., dogs and cats) animals have frequently been reported to attack olive ridley sea turtles. This happens mostly by the beach or along the coast when turtle approach land to lay their eggs [96,97,98]. The consequences range from minor injuries to the death of individuals. For example, attacks by dogs and other semi-domesticated animals are a common phenomenon, while in Costa Rica, jaguar (*Panthera onca*) attacks on greens and olive ridley turtles (Figure 3) are commonly observed [99].

Sharks, killer whales and saltwater crocodiles are possibly olive ridley sea turtle’s only natural predators after they reach adulthood [100,101]. Opportunistic predation of olive ridley turtle eggs and hatchlings by non-native and native predators (particularly feral pigs, coyotes, coatis, birds, and crabs) are well known [45,102,103], and it is usually common in nests that have been exposed due to beach erosion or dogs [96]. As a logical result, this can have negative effects on the hatching success rate. Moreover, depredation of olive ridley turtle nests by ants (*Dorylus orientalis*) have been documented resulting in 0% hatching success [104], whereas the sapro-necrophagous beetle *Omorgus suberosus* that feeds on live and decomposing eggs poses also a serious threat to olive ridley nests in La Escobilla nesting beach [105]. Additionally, the high abundance and proliferation of bacteria and fungi results in a lack of oxygen that affects hatching success on the sand at the beaches, which could be causing the decline of populations on arribada beaches such as Nancite, Costa Rica [60,62]. 

### 3.4. Infectious Diseases 

Fibropapillomatosis (FP) is one of the most significant transmissible diseases known in sea turtles and remains a persistent health concern despite conservation successes and significant growth of some affected populations [36,106]. FP is associated with Chelonid alphaherpesvirus 5 (ChHV5) and was first observed in olive ridley turtles in Costa Rica in 1982 at Ostional National Wildlife Refuge [107,108]. Aguirre et al. [109] confirmed the characteristics of the disease in olive ridley tumours from Ostional NWR by histopathology. Subsequently, ChHV5 was detected in diseased and healthy tissues from olive ridley turtles [110,111] and in completely asymptomatic animals from the same nesting site [36,112]. Fibropapilloma tumours have been also observed in olive ridleys nesting on the Pacific coasts of Costa Rica [109,113], in “La Escobilla” sanctuary in Oaxaca, Mexico [114], in Nicaragua [115], in India [116], and foraging off the Pacific coast of Mexico [117,118]. Recently, FP has been reported in olive ridleys stranded in Chile [119] and in the U.S. [106,120]. Reports and sightings of olive ridley turtles with FP in Pacific and Atlantic waters, as well as nesting beaches, are becoming more and more frequent. Although there are no current epidemiological data to confirm its impact on these species’ populations, FP could be considered a threat for the decline of olive ridley populations and an indication of range expansion. Nevertheless, it may be relevant to mention that there is also evidence suggesting a potential combination of genetic (either from the host and the virus) and environmental factors that may prevent some turtle populations across species from being afflicted by the disease [36].

### 3.5. Ingestion of Marine Debris 

Olive ridley turtles may consume human-discarded marine waste such as fishing lines, balloons, plastic bags, floating tar or oil, and other objects that they mistake for food [95]. A floating plastic bag can resemble a jellyfish, algae, or other organisms that are common in sea turtle diets. In fact, loggerhead turtles ate plastic 17% of the time they came upon it, mistaking it for jellyfish. Green turtles, who are likely on the lookout for algae, increased this ratio to 62%. Marine debris has been identified in the stomachs of dead olive ridleys as well as in the nostrils of live olive ridley turtles in Costa Rica [121,122]. All sea turtle hatchlings and juveniles are particularly vulnerable to becoming entangled in rubbish in the oceans and on the beach, including old plastic robes, ‘six pack’ holders and discarded fishing gear (Figure 4).

Consumption of synthetic materials disrupts the metabolism and causes harm to sea turtles by allowing harmful and toxic chemical compounds to enter the body [123]. Sharp plastics can rupture internal organs, and bags can induce intestinal blockages, leaving turtles starving and unable to feed. Even if they survive, eaten plastic may cause turtles to become excessively buoyant, stunting their growth and resulting in low reproduction rates [95].

Pesticides, heavy metals, and PCBs have all been found in turtles and eggs. These inorganic elements are absorbed by organisms by direct exposure, through the diet, or vertically [124,125]. Additionally, an association between inorganic elements and the hormone corticosterone have been observed in olive ridley turtles, which suggests that these could be generating stress responses in these individuals that previously had ingested some sort of marine debris [126]. The presence of non-essential trace elements such as Cd in olive ridley eggs could have negative effects on the turtle’s health and on people who regularly consume eggs, mainly children [127]. However, more studies are needed to clarify these issues. 

### 3.6. Marine Oil Pollution

Oil pollution of near shore and offshore marine environments has been for decades and remains now a threat to all sea turtles, including eggs and hatchlings. Oil spills wreak havoc on the lungs, skin, blood, and salt glands of marine animals affected including sea turtles [128]. In addition, feeding areas may overlap with mining and oil or gas exploration activities, posing a potential threat to turtle health [72].

### 3.7. Ship Container Strikes 

Throughout their range, sea turtles are threatened by vessel collisions at ports and waterways along developed coastlines, and even during their oceanic migrations in the open sea. When sea turtles are at or near the surface, several forms of watercraft might strike them, causing injury or death [129]. Areas with a lot of boats, including marinas and inlets, are more dangerous. When making reproductive migrations and while near shore during the nesting season, adult sea turtles, particularly nesting females, are more vulnerable to vessel strikes [130]. 

### 3.8. Climate Change, High Temperatures, Sea Level Rise and Coastal Development

The olive ridley’s habitat and biology is already being threatened and will be further harmed by climate change [131]. A warming environment has been anticipated to cause changes in beach morphology and to generate higher sand temperatures, which could be fatal to the eggs produced, and ultimately would modify the ratio of male and female sea turtle hatchlings seasonally generated [86]. Sand temperature increases can result in the birth of exclusively females (sex is determinate in sea turtles by the egg’s incubation temperature in the sand). Storms induce beach erosion, which can cause nests to flood or be washed away. Changes in the amount and distribution of food supplies in the maritime environment are expected to modify the migratory and foraging ranges, as well as the nesting season of olive ridleys [86]. Sea level rise, which can destroy nesting beaches and induce coral bleaching, a critical habitat for sea turtles (in particular for hawksbill turtle species, *Eretmochelys imbricata*), is also a result of climate change [132]. One of the main direct consequences of the current trend of climate change is the rise in global temperatures in the environment. If the temperature increases, it may affect the recovery and survival status of sea turtle populations. As previously mentioned, sex determination in sea turtles is temperature sensitive, and the warmer the incubation temperatures are, the more females than males will be generated [133]. The pivotal temperature (PT) is defined as the temperature that produces the same proportion of males and females, and can vary from one population to another, or even within the same turtle species. For example, in Brazil, the PT for olive ridley turtle nest incubation was estimated to be 30.7 °C, in Costa Rica it was 30.5 °C, and in Mexico the PT has been calculated to be 29.9 °C [134,135,136,137]. Sea turtles’ sex ratio is projected to tilt to more females than males as a result of rising temperatures linked to global warming. The sex of a sea turtle is decided during the third incubation stage, when the embryo’s sex is sensitive to temperature changes [138]. As such, hatchlings will be predominantly males if the temperatures are below 29 °C, and predominantly females in the range 30–31 °C and over 31 °C [136]. The upper fatal limit for embryo development is 35 °C, where hatching success is as low as 2% [134]. In Ostional, Costa Rica, it has been recorded that the embryo lethal effect due to high temperatures occurs mainly during the dry season, due to the absence of the cooling effect of rain [134]. There is a high possibility that greater temperatures will have a significant negative impact on nest hatchling survival, sex ratio, and also possibly genetic diversity loss, as a direct result of the lower male production directed linked to the higher temperatures. Prolonged exposure to high incubation temperatures can additionally result in decreased emergence success as well as reduced hatchling quality due to smaller carapace size and decreased locomotors performance, increasing the likelihood of predation [139]. In Costa Rica, studies have determined that on solitary nesting beaches the incubation temperature is close to the pivotal temperature; in addition, a greater hatching success compared to arribada beaches was reported, probably because the temperature inside the nest increases less due to a lower amount of organic matter [140]. In this way, solitary nesting beaches could be generating a higher proportion of male turtles, which highlights the importance of establishing conservation strategies in these sites.

As sea levels will rise as a direct result of the eminent global warming, there will be a substantial loss of the natural environments where sea turtles lay their eggs along the coastlines on tropical beaches [131]. Sea turtles usually lay their eggs between the mid-beach slope and the vegetation line, a short distance from the ocean waves. As sea levels rise, extra moisture from wave action/surf will disturb the nests, exposing the eggs to predators or direct sunlight [141]. If current CO_2_ emissions continue, then by increasing the greenhouse effect in the atmosphere, sea levels are anticipated to rise from 8 inches to more than 6 feet by 2100 [142]. Some coastal communities, particularly tourist destinations across the tropics, are planning to fight sea level rise by building sea walls to protect themselves from the effects of waves. For example, many hotels in Puerto Vallarta, Mexico, have already started erecting barriers to defend themselves [143]. A large number of communities across many other countries are following the same line of background idea to prevent the potential consequences of sea level rise. The construction of sea walls will remove appropriate nesting habitats and cause beach erosion [131]. 

The sea turtle’s ability to nest and feed can be harmed by construction on nesting beaches, mangroves, and other vital coastal environments, and hatchlings may not survive [88,144]. A decrease in the number of nesting olive ridley females was reported in some of the main locations across India due to the reduction in the nesting beach area, where erosion and inundation caused the loss of nests [45]. Coastal development also poses a threat to freshly hatched turtles due to the impacts of light pollution [145]. Hatchlings that may orient themselves to the sea via light reflection signals in the ocean are now misled into going towards land, where they die of thirst or weariness, or are killed on roadways [86]. 

### 3.9. Oceanographic and Other Climatic Events

Regions historically known to have high climatic and oceanographic heterogeneity events can pose a risk to sea turtle populations. Among these events, El Niño Southern Oscillation (ENSO) has had some negative effects on olive ridley turtles populations along the Pacific Ocean. The ENSO is a periodic fluctuation in sea surface temperature (El Niño) and the air pressure of the overlying atmosphere (Southern Oscillation) across the equatorial Pacific Ocean. El Niño occurs every two to seven years and can last anywhere from a year to many years. El Niño causes higher oceanic temperatures in the central equatorial Pacific and is linked to decreased marine ecosystem production [146]. Olive ridley turtles are affected by poor production because they must travel to different places to forage. Furthermore, as a result of the increased temperatures and lower productivity, olive ridley turtles have reduced the number of female turtles that nest [147]. Although changes in their migration patterns and effects on the number of eggs lay by females have also been observed, olive ridley turtles appear to be resilient to this phenomenon and quickly recover [18]. However, long-term monitoring is needed to elucidate its effects on the abundance of olive ridley turtles [85]. 

Other events such as hurricanes, storms and floods can cause erosion of nesting beaches, with the complete loss of nests as a consequence [148].

### 3.10. Harmful Algal Blooms

Harmful algal blooms (HAB) are proliferations of one or different toxin-producing microorganisms that grow excessively in a body of water, symptomatic of ecosystem imbalance [149]. Sea turtles are exposed to these natural toxins mainly by dietary exposure [150]. Particularly, dinoflagellates such as *Gymnodinium catenatum*, *Pyrodinium bahamense*, and *Karenia brevis* produce neurotoxins known as saxitoxins and brevetoxins, which appear to be a threat to the sea turtle’s health and survival. These neurotoxins can have various effects such as increased stranding incidence, altered immune function, affected neurological function, and mortality [24,150]. It has also been proposed and speculated that these neurotoxins could be a tumour promoter [151]. There have been several reports of olive ridley turtles from the Mexican Pacific exhibiting clinical signs of Paralytic Shellfish Syndrome (produced by saxitoxins) in Bahía Banderas and in Oaxaca coast. Affected turtles were lethargic, exhibited impaired motor coordination, demonstrated difficulty in breathing and submerging, and some died [23,24]. It has been hypothesized that a chronic intake of biotoxins is the cause of muscle weakness of the posterior flippers (paresis) in nesting olive ridley turtles of La Escobilla, which represents a limitation for the construction of nests [152].

## 4. Conservation Efforts and Management Status 

Despite the fact that the olive ridley is widely regarded as the most abundant of the seven sea turtle species, it is also under serious threat. According to all estimations, olive ridley population numbers have decreased dramatically compared to previous assessments [86]. Their precarious status is mostly due to over-harvesting of their eggs in the previous century and to the hunting for the female’s skin and flesh at nesting locations [107].

Since the olive ridley turtle population has declined over the last thirty years, international treaties and agreements as well as national laws currently protect this species. Currently, olive ridleys are listed as vulnerable according to the red list of threatened species of [3], and are included in Appendix I of the Convention on International Trade in Endangered Species [153]. Moreover, certain olive ridley populations are federally designated as threatened or endangered [107,154]. In the U.S., NOAA Fisheries and the U.S. Fish and Wildlife Service (FWS) have been working together since 1977 to officially protect olive ridleys. Sea turtles designated under the Endangered Species Act (ESA) are under the jurisdiction of both the Fish and Wildlife Service and the Environmental Protection Agency. On the one hand, NOAA Fisheries is in charge of sea turtle conservation and recovery in the maritime environment, while on the other, the U.S. FWS are in charge of sea turtle nesting beach conservation and rehabilitation [86]. Additionally, olive ridley populations are also monitored on a regular basis by NOAA’s National Marine Fisheries Service and the U.S. FWS [107].

A lot of local communities and localities rely on ecotourism activities such as turtle watching or diving for jobs and revenue [155]. Seeing a sea turtle in the wild also has emotional and psychological benefits, so in addition to the economic benefits, ecotourism is a way to promote the conservation of these species, as long as it is carried out in a regulated manner [156,157]. Beach initiatives take advantage of sea turtles as a flagship species and provide possibilities to conserve them, in addition to their relevance for conservation, research, and education. Many countries, including Costa Rica, Mexico, Brazil, Colombia, Guatemala, and Kenya, have community-based conservation projects in nesting beaches [85]. Moreover, schools and universities and other organizations (e.g., Boy Scouts) organize visits for students and volunteers to participate in conservation projects. At some nesting sites in Mexico, Indonesia and the Central Coast of Africa, evaluations of conservation efforts have shown a positive impact on olive ridleys, while in some others, the trends are stable or continue to decline [26,84,158,159]. However, due to the lack of evaluation of conservation efforts at many of the nesting beaches, there is still no understanding of nesting trends or priority areas for intensive management of this species [157]. Sea turtle tourism, such as observation at sea and on nesting beaches, is also on the rise, providing local communities with a steady source of cash. Additionally, the benefits of tourism may pressure governments to increase efforts to mitigate threats to olive ridley turtles as well as the application of legal sanctions, and might also justify the creation of reserves and protected national parks [155,160]. Some conservationists have suggested that similar tactics can be applied to other arribada areas with stable or expanding populations, although this concept is divisive [161].

In certain underprivileged coastal communities, the amount of eggs laid at mass-nesting sites provides food and cash for the locals. The legal, communal egg collection program at the Ostional National Wildlife Refuge in Costa Rica has been largely effective in the 40 years since it was established, with long-term monitoring studies suggesting that the rookery nesting there is stable [65]. Furthermore, studies on the black-market egg trade suggest that these eggs could help to suffocate the black-market egg trade. The community egg harvest initiative continues to generate significant funds and resources for conservation as well as to help local families make ends meet.

On the other hand, fishing gear improvements (such as the adoption of Turtle Exclusion Devices, TEDs), adjustments to fishing methods, and closures of particular regions to fishing during nesting and hatching seasons are initiatives that are helping to reduce bycatch of olive ridley unintentional capture by commercial and sport fisherman [107]. Some countries such as the U.S. have approved legislation requiring all shrimp sold in the country to be collected by enterprises that use TEDs, which allows sea turtles to safely avoid being caught in shrimping nets. 

In addition, the creation of Regional Management Units (RMUs) for olive ridley turtle populations is another important conservation strategy. The implementation of the RMU has contributed to the identification of essential habitats that could merit a greater investment in conservation resources, as well as the evaluation of the threats and the conservation status of this species at the RMU level [29]. 

Despite the fact that olive ridley nesting beaches are protected by many nations, eggs are still taken, and nesting females are slain for their flesh and skin. In addition, and although many commercial fishing fleets now use TEDs in their nets, fishing nets also take a heavy toll, snagging and drowning these turtles on a regular basis [19,162]. Therefore, olive ridley worldwide populations will continue to decline unless poaching laws are better enforced and the universal use of turtle excluder devices is implemented in trawl nets fishing vessels, mainly in the most important feeding and breeding sites of this species [163]. 

### Perspectives for the Protections of the Olive Ridley Turtle 

There are several aspects to be considered for the protection and conservation of olive ridley sea turtles. Next, we bullet point some of the most important objectives to increase the generation of knowledge, to promote and to encourage for the conservation of these chelonids:

To create cooperation and collaboration with international stakeholders to put conservation measures in place and create agreements, such as international treaties to conserve sea turtles;

To promote the creation of national parks and nature reserves for the protection of nature sites for this species;

To support the collection of robust data on the bycatch rate to generate gear recommendations, restrictions or modifications in the areas that best suit both the species and the fishermen;

To encourage changes in fishing gear practices and/or fishing gear modifications (e.g., TEDs), using large circle hooks in longline fisheries, and implementing spatial or temporal closures to avoid or minimize bycatch are all being researched, developed, and implemented;

To promote smart and clever fishing practices via education. Sea turtles can be entangled and killed by hooks, lines, or nets left in the ocean. Thus, to reduce injuries, the use of barbless circular hooks and knot-free buoy lines should be promoted;

To implement vessel-based or airborne surveys, nesting beach studies, satellite tracking, genomics, and mark-recapture (flipper tagging) investigations to keep tabs on populations;

To create support for the long-term research of the biology of olive ridleys. Individuals sea turtle should be tracked throughout time in order to gain a better understanding of essential aspects of their lives, such as growth and maturation;

To properly address the climate change impacts as well as changes in environmental and ocean conditions, which have an impact on sea turtle quantity, distribution, and demographics;

To evaluate the different conservation strategies at nesting beaches by analysing nesting trends and estimating population abundance;

To adjust funding and conditions accordingly. On nesting beaches, where financial funds are limited, focus on protection and patrolling of small beaches and the most productive parts of large beaches;

To conduct threat research and designing conservation strategies to mitigate dangers and encourage recovery;

To establish ecotourism programs as an economic alternative to replace and eliminate the use of olive ridley turtle products (e.g., eggs and meat);

To apply and implement community-based conservation strategies, mainly on nesting beaches;

To incorporate social sciences in the study and evaluation of sea turtle conservation strategies to update legal and socio-economical frameworks;

To collect data on the biology, ecology and health of the species in order to better inform conservation management plans and monitor recovery efforts;

To use stranding and fisheries bycatch databases to assess life history and population health simultaneously with migration patterns;

To understand demography, physiology, habitat utilization, and resource requirements through studying foraging and reproductive behaviour;

To build capacity and raising knowledge of risks to sea turtles, emphasizing the importance of sea turtle conservation, and sharing ways that people can help sea turtles are all goals of this project;

To train and provide the most up-to-date scientific approaches and tools for tracking sea turtle populations around the world; 

To ensure the collaboration with partners to research and raise awareness of the illegal trade of sea turtles;

To guarantee to the coming generations that these animals can be saved and preserved through education and environmentally oriented actions.

## 5. Conclusions

The olive ridley sea turtle is a widespread cosmopolitan species, yet they are still a mystery, and their populations are threatened due to human activities. Their arribadas influence not just global trends and status, but also the public’s perception of turtles and oceanic wildlife in general. Olive ridley turtles are a commodity for non-consumptive uses such as tourism, educational and scientific research, job creation and information services, and a variety of other economic benefits. Sea turtles have historically been utilized for food and the creation of other by-products such as bone, leather, oil, and shell from certain portions of the body since time immemorial. These marine reptiles are a unique piece of a complex ecosystem whose survival is dependent on both exploitable products and potential ecosystem services. These creatures have tremendous cultural, historical and economical significance. These reptiles also play a vital role in industrialized cultures, making them excellent for educational and scientific operations. If we are to ensure a safe future for sea turtles, we must put in place long-term, effective policies wherever they appear.

## Figures and Tables

**Figure 1 animals-12-01837-f001:**
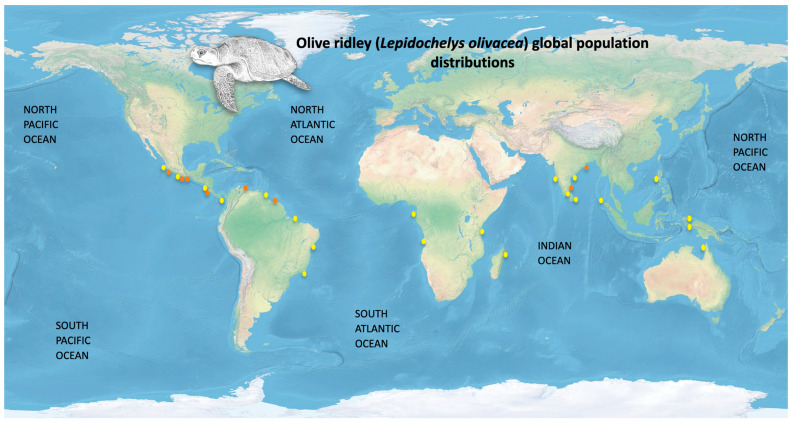
Olive ridley sea turtle (*L. olivacea*) distribution map: Orange circles are major nesting grounds; yellow circles are minor nesting sites (Image taken and modified from [36] with data collected from [5]).

**Figure 2 animals-12-01837-f002:**
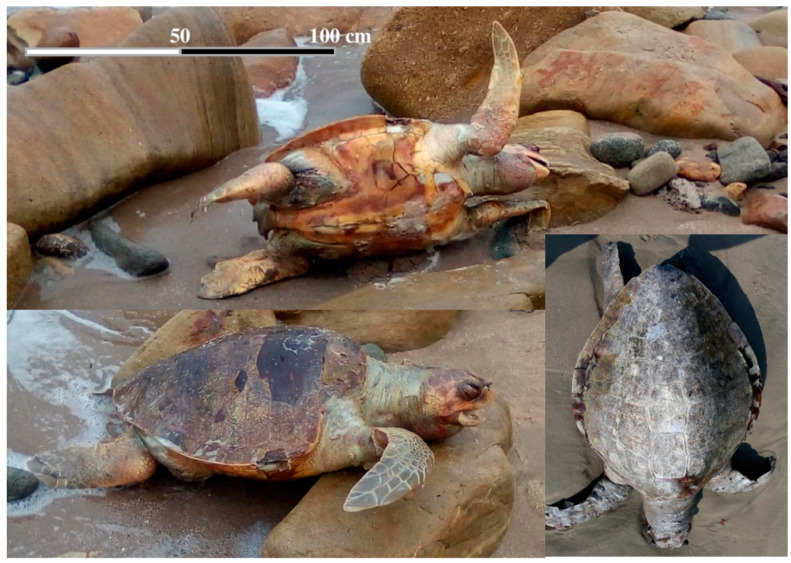
Bycatch and death of olive ridley (*L. olivacea*) sea turtles during fishing activities reported in Bahía de Caráquez-Ecuador (Photo by Lenin Cáceres-Farias).

**Figure 3 animals-12-01837-f003:**
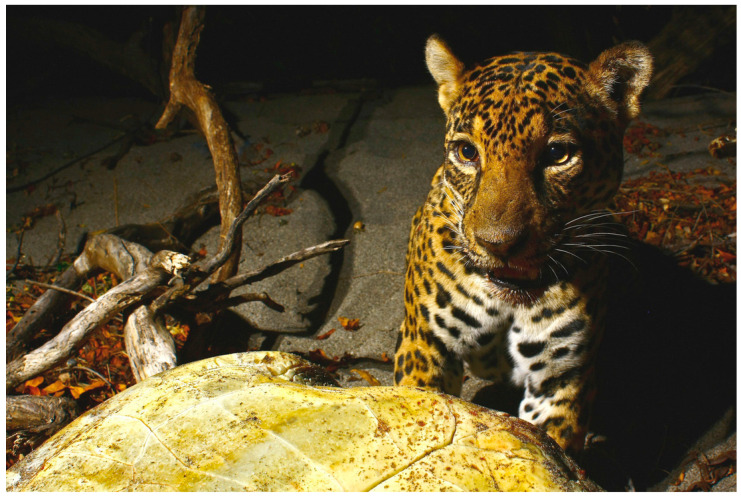
Documented olive ridley turtle (*L. olivacea*) kill by a jaguar (*Panthera onca*) at Santa Rosa National Park, Costa Rica (Photo by Alonso Sánchez from ACR Wildlife Photography, San José, Costa Rica 2022).

**Figure 4 animals-12-01837-f004:**
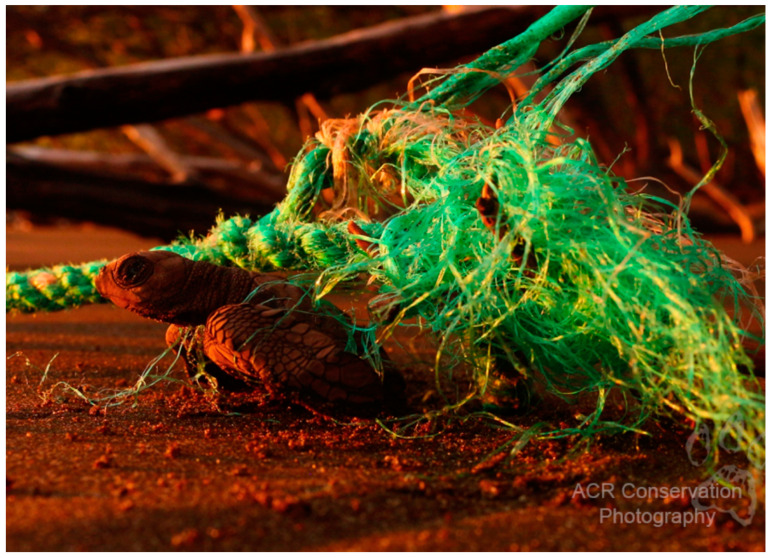
Adapted photo with permission (Photo by Alonso Sánchez from ACR Wildlife Photography, San José, Costa Rica 2022) of a documented olive ridley hatchling (*L. olivacea*) entangled in an old plastic robe washed upon the beach in Nancite, Costa Rica.

## Data Availability

Not applicable.

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
