# Peer review of "Threats and Vulnerabilities for the Globally Distributed Olive Ridley (Lepidochelys olivacea) Sea Turtle: A Historical and Current Status Evaluation"

_animals, 2022, doi:10.3390/ani12141837_

Round 1

Reviewer 1 Report

Dear Authors,

This is an interesting review but I feel that you need to do a better job integrating the literature you cite.  Some of your comments are poorly supported by the literature you cite, suggesting that you have only a peripheral understanding of the subject.  I also think that although Pamala Plotkin is a significant contributor to the olive ridley sea turtle biology, you need to try to expand your reading to go beyond the obvious, and integrate some great work beyond Dr. Plotkin and on-line publications, and synthesize meaning to what you are writing by genuinely bridging your thoughts with the data in the liteature.  Even though you are submitting a review, I was not clear what the thesis of the review is?  Where is the historical evaluation?  This wasn't very clear to me, and less clear was how you used available knowledge to craft a current evaluation of the population?  Lot's of cited papers, lots of interesting facts- but the limited depth forces me to question your depth of understanding on the topic you are reviewing. My suggestion is to focus this review a bit so you can develop more linear ideas.

Reviewer 2 Report

Congratulations to the authors because the work is very well written with many bibliographical references and there is a fairly extensive work on the current threats to this species of sea turtle. On the other hand, many conservation proposals are offered so that in the future, countries and organizations can act in the different places where these animals live.

The introduction is appropriately sized and they do an excellent job of reviewing other articles on the species. Perhaps he would have put some information comparing his morphology with other species of sea turtle.

The photographs are too dark and it is a bit difficult to see the points where these animals live.

I find it very interesting to analyze point by point the threats suffered by these animals and I consider that the size of the text is adequate because otherwise the work would be very extensive. It is evident that with all the information that is published about it, a book could be written and the authors have very well summarized all the information in the work.

I consider that for the understanding of the work, some graphs and charts should have been introduced that would have made the work more understandable.

In short, it seems to me an excellent job but I consider it appropriate to improve the images and put some graph or table

Reviewer 3 Report

This is an interesting manuscript (presented as Review) on the threats and vulnerabilities for the olive ridley sea turtle (Lepidochelys olivacea), a very interesting topic, taking into account the conservation status of their populations around the world. It is remarkable the comprehensive review made by the authors including around 165 references, many of them very recent. Also noteworthy are the contributions of the authors in the ‘Perspectives for the protection of the olive ridley turtle’ section. However, in my opinion, the article needs a major revision.

My main criticism is the management that the authors have made of the included references. As I mentioned before, the manuscript stands out for the number of references included, but the authors should have spent more time checking that all the references included in the text are also included in the list of references and vice versa.

The following references that are cited in the text are not found in the reference list:

WWF 2021 (Line 118, L271)

Hudgins et al 2017 (L133, L251)

MTSG 2003 (L275)

The following references included in the reference list are not found in the text:

Abreu-Grobois and Plotkin 2007

Aguirre et al 2002

Chan 2001

Comisión Nacional de Áreas Naturales Protegidas (and year?)

Dash and Kar 1990

Hamann et al 2006

Kurniawan and Gitayana 2020

Limpus 2008

MTSG 1995

Nahill 2021

Olive ridley Project ORP 2020

Phillott and Rees 2020

Reichart 1993

Thorbjarnarson et al 2000

Ulaiwi 1997

I have other suggestions and/or doubts about some references:

L119: Is the reference ‘Seminoff et al 2014’ correct? (the title is about loggerhead sea turtle)

L151: Is ‘Barrietos-Munoz et al 2014’ or ‘Barrientos-Munoz et al…’?

L185-186: Sardeshpande and MacMillan 2018 or 2019? In the reference list it’s 2019

L461: Cortés-Gómez et al 2017 or 2018? In the reference list it’s 2018

L745-748: Are repeated these two references?

L814: Chaves Ramirez (2017a): delete ‘a’

L817: Chaves et al (2017b): delete ‘b’

L863: please, separate the two references

L999: United States Fish and Wildlife Service (capitalized initials)

L1057: Reséndiz et al 2015: what is the title of the paper?

L1134-1137: These two references are repeated. Note that with the next reference (lines L1138-1139) there will then only be two references Whiting et al 2007; therefore only 'a' and 'b'

Therefore, I suggest the authors check all the references and correct all the detected deficiencies. Please note that this reviewer may have missed some additional errors.

I also urge authors to follow the proper format for references in the new version of the manuscript. In the text, references must be numbered (reference numbers placed in square brackets) in order of appearance (please, see the Instructions for Authors) and listed individually in the References section.

Abstract

The authors have only included an Abstract in the manuscript. However, according to the instructions for authors, they should also include a Simple Summary (please, see Instructions for Authors to see differences). 

Introduction

In my opinion, the section of the text that requires the most changes is the Introduction. I consider that it is too long, somewhat messy, and also many of the statements are repeated later throughout the text. Authors should shorten it significantly.

General characteristics

Line L108: ‘its smaller sister’: this is not a suitable expression for a scientific review.  

Habitat, food and feeding behavior

Lines L264-265: this sentence about marine litter needs a reference. 

Source of food for human

Lines L285-290. This paragraph is not related to source of food for human. I suggest moving it to line L622 section ‘Conservation efforts & management status’.

Perspectives for the protection of the olive ridley turtle

I consider it a good format to list the ideas contributed by the authors as bullet points. But please make bullet points visible.

I also suggest the authors group the different bullet points according to the topic to be addressed, in order to facilitate reading. In my opinion, they are currently a bit messy.

And finally, taking into account the title of the special issue 'Veterinary Sciences and Sea Turtles' in which this review is intended to be included, the authors should add some bullet points addressing the role of veterinary sciences in the conservation of this species of sea turtle.

Declaration of Interest Statement

In my opinion (although it must be the editor who approves it) the sentences about Mr Md Simul Bhuyan should be deleted (Lines L731-734).

Round 2

Reviewer 3 Report

Thanks for your efforts to improve the manuscript. My main criticism about the first version of the manuscript was the management of the references.  Given the large number of bibliographical references included, the existence of errors is understandable, a fact that has been corrected in this second version.

Regarding my other suggestions, some have been taken into account and others have been clarified by the authors in the rebuttal letter, so I believe that the manuscript can be accepted for publication.